# ViML: A Video, Music, Language Unified Dataset for Understanding and Generation

## Abstract

Integrating multimodal understanding and generation into a unified framework can bridge the domain gap across different modalities. However, existing multimodal-language datasets predominantly offer text descriptions for a single modality, treating visual and audio as separate tasks. This approach neglects the inherent audio-visual correlations, resulting in annotations that are often monotonous and modality-specific rather than comprehensive and precise. Such oversight hampers the advancement of cross-modality research. To fulfill this gap, we present ViML, a large-scale multi-modality-to-language dataset incorporating 3M video clips with high-quality multimodal captions. In ViML, we propose a systemic captioning framework, achieving various modality annotations with more than 12.2k hours of trailer videos. Here, to ensure the caption retains music perspective while preserving the authority of visual context, we leverage the advanced LLM to merge all annotations adaptively. In particular, the ViML has two main advantages: (1) the topics are diverse, and the content characters are of various types, *e.g.*, film, news, and gaming. (2) the corresponding background music is custom-designed, making it more coherent with the visual context. In this fashion, our ViML dataset potentially paves the path for fine-grained large multimodal-language model training. In experiments, we provide evaluation metrics and benchmark results on our dataset, demonstrating the high quality of our annotation and its effectiveness for model training. We include demo data in https://anonymous.4open.science/w/ViML-4C78

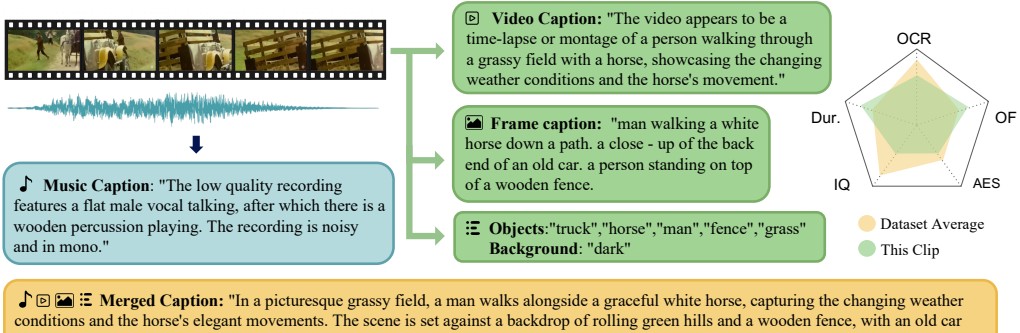

Figure 1: We present a video-language dataset with music captions, **ViML**.

## 1 Introduction

Integrating multimodal understanding and generation into a unified framework can bridge the domain gap across different modalities and has a wide range of applications in daily life, such as AI-driven movies and short video production. Clearly, creating vivid videos requires more than just visual frame generation or individual modality-based approaches; it necessitates an integrated understanding and generation capability. Thanks to various large-scale video-language datasets, numerous generative multimodal large language models have been developed to achieve this goal (Blattmann et al., 2023a; Long et al., 2024; Chen et al., 2023a; He et al., 2023; Lin et al., 2023; Kondratyuk et al., 2023;

Wang et al., 2023a; Henschel et al., 2024). However, existing video-language datasets ( Chen et al. (2024b); Miech et al. (2019); Wang et al. (2023c)) typically focus on visual-based text descriptions, overlooking the significance of inherent visual-audio dependencies. This presents a complex challenge that demands cohesive integration of multiple modalities, yet remains largely unexplored.

Such ignorance might because collecting high-quality multi-modality source data that preserves consistency between different modalities is extremely challenging. Unlike previous datasets that only provide visual frame-based caption (Bain et al., 2021a), multimodal datasets contain complex data formats (*e.g.*, music), resulting in more labor-intensive and time-consuming costs in data processing and annotation. Moreover, achieving a high correlation between audio and visual content presents challenges.

Targeting to fill the dataset gap by creating a comprehensive and accurate multi-modality visual-audio dataset, we first notice trailers. As a precursor to a full-length work, the video trailer has emerged as a vital tool for artists to showcase and disseminate their creations. These short videos typically combine the most compelling visual shots with carefully selected music, have high cross-modality consistency, and hold significant potential in broader multimodal research. The topics are diverse, and the content characters are of various types, e.g., film, comedy, and gaming, as shown in Fig. 2. Significantly, the trailer format represents a unique, high-quality, video-centric multimodal data source that benefits further multi-modality research exploration and analysis.

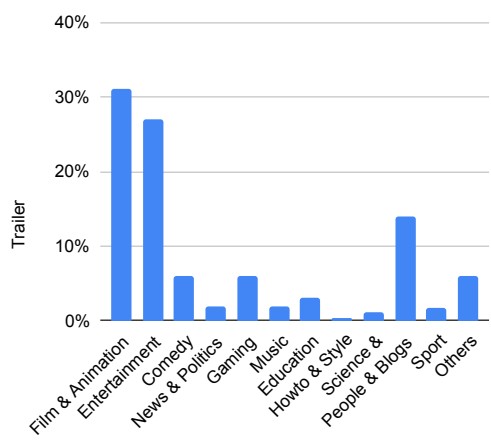

Figure 2: Distribution of video categories of ViML dataset.

In this work, we propose **ViML**, which aims to unlock the potential of multimodal content understanding and generation for innovative applications in video content generation, as displayed in Fig. 1. We first recognize the immense value of trailers as a video-centric dataset, especially considering the music alongside the videos. **ViML** contains 20M+ video clips from 290k trailer videos encompassing various source categories as shown in Fig. 2. To ensure the quality of our dataset, we have carefully designed a robust data filtering and cleaning methodology. We also provide extensive statistics works to demonstrate the diversity and complexity of our dataset.

To address the multimodal to language annotation challenge, we have designed a multimodal captioning pipeline incorporating diverse state-of-the-art (SOTA) captioning models (Doh et al., 2023b; Yu et al., 2022; Liu et al., 2024). Furthermore, we propose a language model fusion strategy to generate fine-grained multimodal captions. We have performed small-scale annotations on the entire dataset, created a multimodal annotation subset of 3 million samples **ViML-3M**, and provided a testing set **ViML-Test** with manually-adjusted multimodal caption.

We present evaluation metrics and benchmark results on our dataset, demonstrating the high quality of our annotations and their effectiveness for model training. Through extensive experiments and benchmarking, we showcase the difficulty and diversity of our dataset using various evaluation metrics. We also conduct human evaluations to validate the quality of our multimodal captioning pipeline. Furthermore, we fine-tune understanding models (Zhang et al., 2023a), video-to-music model (Tian et al., 2024), and generative models (Chen et al., 2024a) on a subset of our dataset, providing evidence of its high quality and efficacy. Additionally, we evaluate video understanding models on the ViML-Test, and MSR-VTT Xu et al. (2016), highlighting the challenges posed by our dataset, and evaluate video-music-based models to demonstrate the effectiveness of cross-modality tasks.

Table 1: Comparison of ViML-X and other Video to language datasets. ViML-X contains three sets(20M,3M, test) with 720p resolution.

| Dataset | Year | Size | Caption | Modality | Clips | E(V) | E(T) | Resolution |
|---|---|---|---|---|---|---|---|---|
| WebVid (Bain et al., 2021a) | 2021 | 52khr | Alt-text | Video | 10M | 10s | - | 360p |
| Panda (Chen et al., 2024b) | 2024 | 167khr | Auto | Video | 70M | 8.5s | 13.2 | 720p |
| HD-VILA (Xue et al., 2021) | 2022 | 371.5khr | ASR | Video | 100M | 3.6s | 32.5 | 720p |
| MSR-VTT (Xu et al., 2016) | 2016 | 40hr | Manual | Video | 10K | 15s | 9.3 | 240p |
| InternVid (Wang et al., 2023c) | 2023 | 760.3khr | Auto | MM | 100M | 11.7s | 11.6 | 720p |
| HowTo100M (Miech et al., 2019) | 2023 | 134.5khr | ASR | MM | 136M | 3.6s | 4 | 720p |
| **ViML-20M** | 2024 | 27.1khr | Auto | Video | 20M | 4.6s | 10.7 | 720p |
| **ViML-3M** | 2024 | 12.2khr | Auto | MM | 3M | 13.8s | 39.4 | 720p |
| **ViML-Test** | 2024 | 3.2hr | Manual | MM | 1k | 11.6s | 98.2 | 720p |

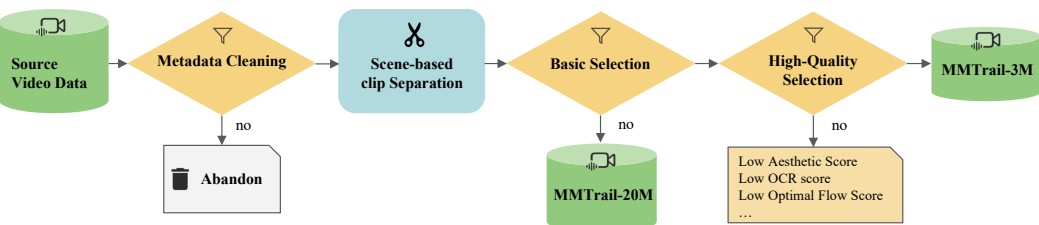

Figure 3: Data collection and cleaning pipeline of the ViML. Starting from the source video data, we follow the metadata cleaning, scene-cut, and basic filtering to obtain the full list of ViML-20M and High-Quality Selection to filter the ViML-3M.

## 2 RELATED WORK

### 2.1 VIDEO GENERATION AND UNDERSTANDING

Video understanding and text-to-video generation are inherently connected tasks. In recent years, there has been remarkable progress in understanding models (Wu et al., 2021; Liu et al., 2021; Zhao et al., 2022; Bain et al., 2021b; Yang et al., 2022b;a; Lin et al., 2022; 2019; Bertasius et al., 2021; Wu et al., 2019; Zhang et al., 2023a; Chen et al., 2023b), which have greatly contributed to the advancement of text-based video generation techniques. The availability of large-scale datasets and diffusion models has revolutionized video generation, moving from pixel-level approaches like (Ho et al., 2022b; Singer et al., 2022; Ho et al., 2022a) to latent-level video diffusion models (He et al., 2022; Zhou et al., 2022; Blattmann et al., 2023b; He et al., 2023). Concurrently, understanding models have also witnessed significant improvements. A series of MLLM-based understanding models (Liu et al., 2024; Maaz et al., 2023; Song et al., 2023; Jin et al., 2023) has reached satisfied understanding abilities. The iterative interaction between video generation and understanding has led to the development of excellent large-scale datasets and models encompassing diverse approaches. Panda (Chen et al., 2024b) introduced an auto-caption model distilled from video understanding models like VideoLlaMA (Zhang et al., 2023a), MiniGPT4 (Zhu et al., 2023b).

### 2.2 VIDEO-LANGUAGE DATASETS

Captioned video datasets are essential for text-to-video generation and understanding tasks. MSR-VTT (Xu et al., 2016), UCF-101 (Soomro et al., 2012) are commonly used as evaluation sets. Anna et al. (Rohrbach et al., 2016) presented 118,081 movie clips with descriptions. ActivityNet Caption (Krishna et al., 2017) by Ranjay et al. is a benchmark involving event detection, natural language description, and event localization. WebVid (Bain et al., 2021a), VideoFactory (Wang et al., 2023b), and other works (Sanabria et al., 2018; Wang et al., 2019; Stroud et al., 2020; Nagrani et al., 2022), contain multilingual video descriptions, video clips, metadata such as titles, descriptions, tags, and channel names, and are used for tasks like video understanding, text-to-video retrieval, and audio-video captioning with weak annotations. Several datasets and approaches have utilized audio to enhance video captioning (Miech et al., 2019; Rohrbach et al., 2016; Zellers et al., 2021; Wang et al., 2023c; Chen et al., 2024b; Xue et al., 2021). These datasets consist of movies, web videos,

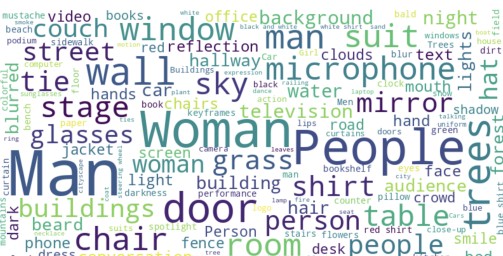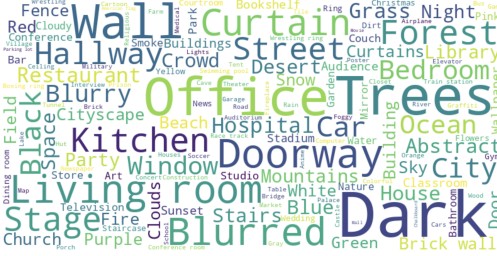

Figure 4: Word cloud of the (left) objects and (right) background in ViML. Most of the objects are human, and most of the backgrounds are indoor scenes like office, kitchen, etc.

YouTube videos, and high-resolution videos from popular YouTube categories, providing transcribed audio descriptions, narrations (Miech et al., 2019), ASR transcriptions (Zellers et al., 2021), and multiple captions generated through a auto caption model (Miech et al., 2019; Chen et al., 2024b; Wang et al., 2023c).

However, existing works only use audio to enhance the video caption, using metadata or ASR to provide extra information for the video caption. A large-scale video-centric dataset that cooperates with high-quality music and multimodal caption is still lacking.

## 3 ViML Dataset

To boost the performance of multimodal generation tasks, we construct a high-quality multimodal dataset based on trailers-like videos, which offers a wealth of multimodal information and diverse categories that distinguish them from existing large-scale video-language datasets. In this section, we introduce an automated data collection and cleaning pipeline to construct the source data for trailers in Section 3.1. We developed a multimodal captioning pipeline that generated rich multimodal captions for video segments in Section 3.2. Additionally, we detail our different subsets in Section 3.3. Moreover, various statistical analysis of ViML Dataset are shown in Fig. 5.

### 3.1 Data Collection Pipeline

Targeting finding the data source for video, music, and language caption, we noticed a keyword: trailers. The trailer videos usually contain multiple themes, including movies, TV shows, games, etc. They are well organized and have the most attractive clips of the corresponding videos. At the same time, most of them are accompanied by corresponding background voices and music.

However, trailers are often vague and incoherent, and the music and voiceover are usually commentaries rather than directly produced by the characters in the picture. To solve this challenging problem, we designed a comprehensive data collection and cleaning process as shown in Fig. 3 to deal with such complex videos, as described in this section. We introduce motion filtering, music detection, OCR filtering, etc., to ensure the high quality of ViML. We include the video examples of each filtering score in web pageshttps://anonymous.4open.science/w/ViML-4C78. The detailed methods are as follows:

**Collection Strategies** We first employ the keyword "trailer" to reselect the existing internet video datasets to increase the dataset collection efficiency. Such general-purpose trailer videos encapsulate a wide range of artistic works and genres. Then, we tailor the keywords more specifically to collect trailer videos with divergent sources explicitly. Those keywords include "Movie Trailers", "Video Game Trailers", "TV Show Trailers", "Documentary Trailers", etc. Together with privacy filtering as described in Section 6, we collect 285,518 comprehensive trailer videos with a total duration of 94,911,802.8 seconds.

**Trimming** To facilitate the extraction of various video information in subsequent analyses, we cut the original videos into clips based on the scenes. As the most mature and practical tool currently available, PySceneDetect[*] offers robust functionalities for this purpose. Thus, we use the

---
[*]https://github.com/Breakthrough/PySceneDetect

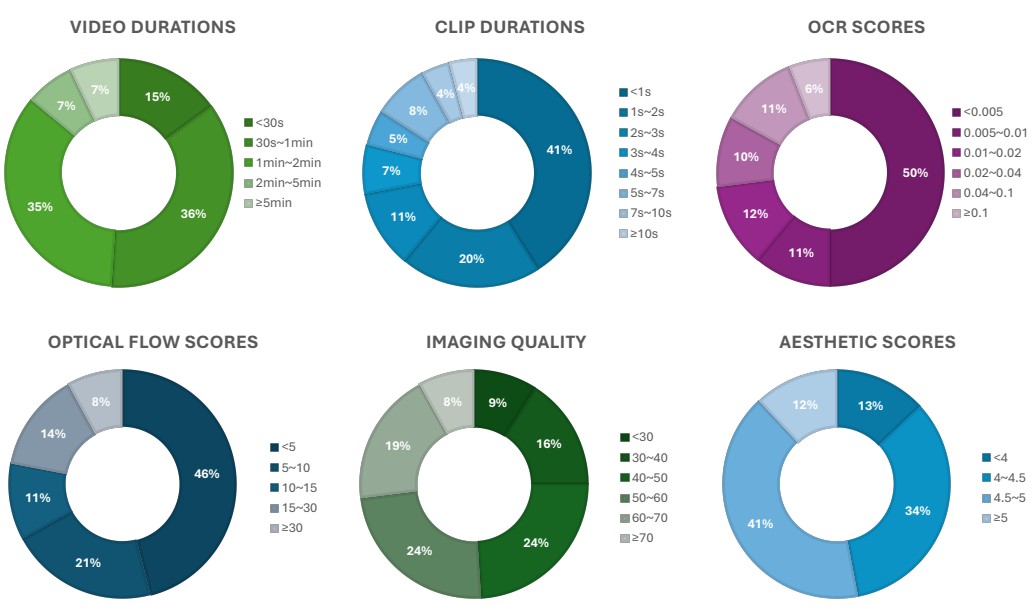

Figure 5: Statistic of the ViML clips. These evaluation scores collectively include OCR score, Video duration, optical flow score, clip duration, image quality, and aesthetic score, demonstrating the richness and diversity of ViML, making it a valuable resource for multimedia research.

PySceneDetect ContentDetector to compare the difference in content between adjacent frames and then cut the videos according to the predetermined threshold of 30. We finally generated 21,588,792 clips with an average duration of 4.6 seconds.

**Motion Filtering** As mentioned, trailer videos are often blurred with large motion, so we applied motion filtering. Motion vectors and optical flow are both mainstream motion quality evaluation methods. In our case, trailer videos often have rapid cuts and transitions between scenes, posing challenges for optical flow-based analysis. Other than optical flow, motion vectors are more robust to these quick changes, as they rely on larger block units' displacement rather than individual pixels' continuous flow. On the other hand, motion vectors are more lightweight and can be calculated more efficiently. Thus, we leverage motion vectors to filter out clips with problems like static frames, title sequences, and slideshow-like playback.

**Diversity** We evaluate the diversity and richness of our dataset from three aspects: theme, objects, and backgrounds. While collecting, we first assess the categories from the Yotoube metadata provided by the video provider, as shown in Fig. 2. Furthermore, we generate an object-level caption list and background by LLaVA (Liu et al., 2024) for a more accurate category-based generation. The word cloud of objects and backgrounds is shown in Fig. 4.

**Music Event Detection** We employ the sound event detection model PANNs Kong et al. (2020) to identify videos containing music events. The model predicts frame-level event labels on the whole dataset, over 70% of the audio segments contain music.

**Audio Video Alignment** We use ImageBind (Girdhar et al., 2023) to assess the semantic alignment between the vision and audio modalities. The model is pre-trained in a CLIP fashion to align six different modalities. ImageBind-AV scores typically indicate a stronger semantic correlation between the vision and audio modalities. We compute the ImageBind-AV scores for all the data to evaluate this alignment.

**OCR** Trailer videos often have text-heavy sections with high-quality text animations, like opening and ending credits. To identify these text-rich segments, we utilize OCR to detect the text content in the video frames and calculate the bounding box area of the text. This measurement reflects the amount of text in the clips.

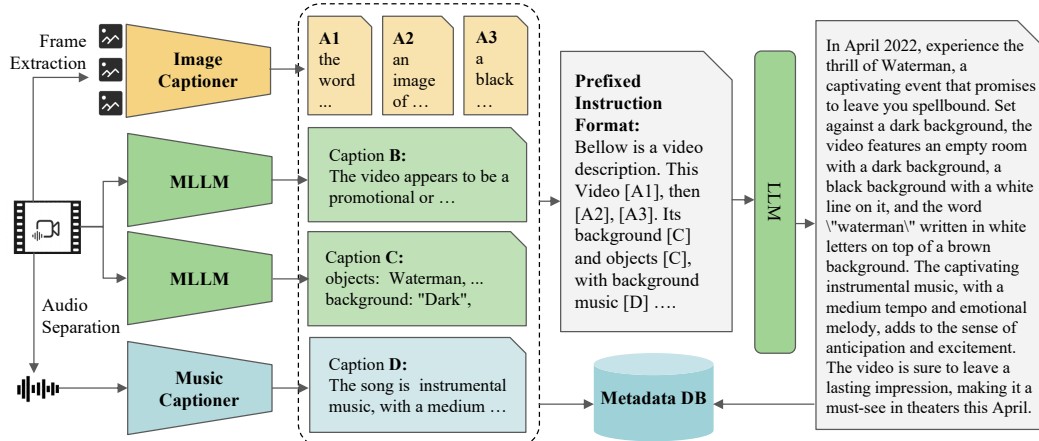

Figure 6: Data captioning pipeline. Starting from video clips, we extract frames and audio and then perform multiple rounds of captioning. A predefined instruction format combines multimodal captions, which serve as prompts for the language model and generate the final merged prompts.

**Quality Statistics** In addition to text detection, we considered image quality (Huang et al., 2024) and aesthetic scores (Schuhmann et al., 2021) to enhance our analysis of videos. These measures allowed us to evaluate frames' visual fidelity, clarity, and aesthetic appeal, providing more comprehensive insights for analysis and editing.

## 3.2 Video Captioning Pipeline

The ViML contains many complex themes, like subtitles and character animations, as shown in Fig. 2, which brings extraordinary complex work for video captioning. At the same time, smooth transition shots also make it impossible for traditional single-frame annotation methods to convey semantics coherently. Therefore, this section introduces a multi-temporal and multimodal caption pipeline containing a detailed video description from frame, motion, and music levels.

**Frame Caption** The auto-captioning pipeline has proven efficient in cutting-edge video generation foundation models. SVD (Blattmann et al., 2023a) and Pandas (Chen et al., 2024b) have given promising results and demonstrated the importance of high-quality frame captions for the generation model. We initially performed image-level captioning on the individual frames of the data. We employed coca (Yu et al., 2022) for each video clip to generate separate captions for three frames(first, middle, and last), resulting in relevant captions.

**Video Caption** Having obtained concise captions for three frames that capture the essential information, we aimed to obtain fine-grained captions and variations between frames in the video. We concatenated multiple frames into a comic strip format and employed the LLaVA (Liu et al., 2024) image model to guide the description of the dynamic differences between frames. Additionally, leveraging a powerful multimodal language model, we incorporated OCR and more detailed summary descriptions to expand the information within the frame captions.

**Categories and Background** Noticing the LLM-based caption has hallucinations when describing the frame, we further generate word-level labels to enhance the annotation of the main objects and background. Initially, we utilized LLaVA's QA capabilities to have the model answer questions about the background. Subsequently, through QA, we prompted the model to provide relevant category information. We conducted the word cloud in Fig. 4. and certified caption quality by subjective experience in Section 4.

**Music Caption** Moreover, given that trailer music usually has a well-designed audio effect and background music, we applied the music caption on our dataset rather than a standard audio caption. In our work, we used MusicCaps (Doh et al., 2023b), an LLM-based music captioning model. The caption format is well designed with its description pipeline, which first describes its sound quality, a generated speech style, and a detailed description of its instrument and music style. More examples

Table 2: Comparison of ViML-3M and other Video-Audio Generation Dataset. For each dataset, we list the following information in each column: dataset name (Dataset), public year (Year), average duration per clip (Dur./Clip), total number of clips (#Clips), total number of hours (#Hours).

| Dataset | Year | Dur./Clip | #Clips | #Hours |
|---|---|---|---|---|
| Audioset (Gemmeke et al., 2017) | 2017 | 10s | 2M | 5.8khr |
| Vggsound (Chen et al., 2020) | 2020 | 10s | 210K | 550hr |
| ViML-3M | 2024 | 13.8s | 3M | 12.2khr |
| ViML-Test | 2024 | 11.6s | 1k | 3.2hr |

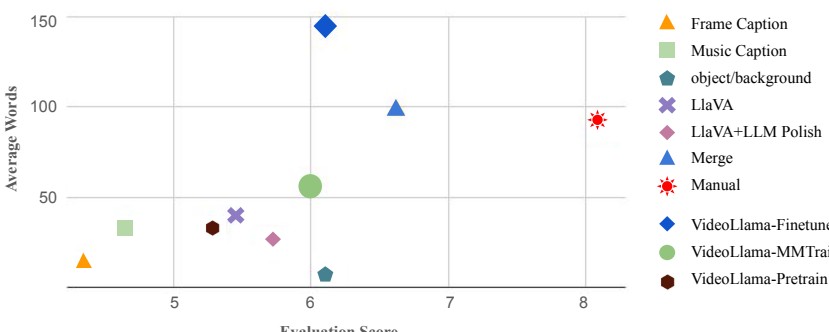

Figure 7: Human evaluation results of the captioning models on the ViML-Test. The X-axis is the average evaluation score from 0-10, and the Y-axis is the average word numbers.

are shown in the Appendix. We further evaluate the generation tasks and the text-to-music generation based on the music caption, which shows the efficiency of our captioning and dataset.

**Merged Caption** Combining all the captions mentioned above, we use the language model llama2-13B (Touvron et al., 2023) to merge all the captions together and generate complete and high-quality multimodal captions. We evaluate the caption accuracy and quality by human preference in Section 4.

### 3.3 SUBSET SEPARATION

We applied the frame caption Section 3.2 for the full ViMLer-20M video clips with 20M+ clips. We included the scale comparison with other large-scale datasets in Table 1, showing that our dataset is a large-scale video-language dataset. ViML has a resolution no smaller than 720p, and ViML-20M clips are 4.6s long on average.

**High-quality Subset** , named ViML-3M, contains a detailed multimodal caption. Compared to the generation distribution, we sampled the top-95% of each eval metric and ensured its diversity of themes to build up a high-quality subset. All clips in ViML-3M are longer than 4s and provided with all the captions, as shown in Fig. 6, including categories, background, frame captions, music captions, merged captions, etc. We also compare ViML-3M with Video-Audio datasets as shown in Table 2, ViML has a larger scale than existing datasets (Audioset (Gemmeke et al., 2017) and Vggsound (Chen et al., 2020)).

**High-quality Testing Set** is extracted from the ViML-3M, we extract a fine-branded testing set that contains 1k video clips and multiple multimodal captions. Then, we manually adjust the merged caption to a manual caption to build a testing subset with trust-wise multimodal prompts. We test several tasks and models on the test set in Section 4 to show the complexity and difficulties of ViML. The test set has 98.2 words of caption on average and includes 3.2hr video clips.

## 4 EXPERIMENTS

This section presents comprehensive experiments on multiple tasks to demonstrate our dataset's effectiveness, diversity, complexity, and difficulty.

Table 3: We compare the data quality of WebVid and ViML, fine-tuning them on the VideoCrafter-2.0 (Chen et al., 2024a) with the same setting on 9 different dimensions. For every dimension, a higher score is better.

| Dimentions(↑) | VideoCrafter-2.0(WebVid) | VideoCrafter-2.0(ViML) |
|---|---|---|
| temporal style | 24.24 | 24.61 |
| appearance style | 24.13 | 24.10 |
| image quality | 63.25 | 69.78 |
| dynamic degree | 41.30 | 43.50 |
| motion smoothness | 96.95 | 98.33 |
| temporal flickering | 98.11 | 98.50 |
| Subject consistency | 96.95 | 98.62 |
| background consistency | 98.42 | 98.40 |
| Overall consistency | 27.33 | 25.33 |
| **Sum** | 63.41 | **64.57** |

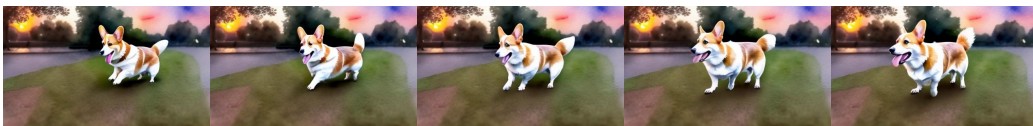

**Caption :** A cute happy Corgi playing in park, sunset, watercolor painting.

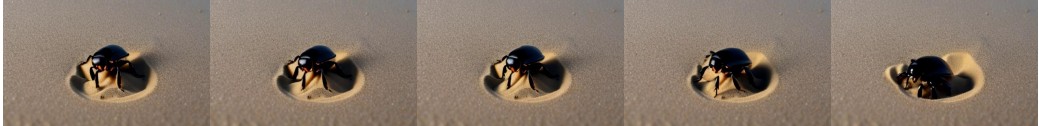

**Caption :** A beetle emerging from the sand.

Figure 8: Two generation result of Videocrafter(ViML). The caption is from the VBench (Huang et al., 2024) evaluation prompts list; the given example shows the high quality in motion and object consistency.

## 4.1 MULTIMODAL CAPTIONING

We present the results of our human evaluation of video caption quality in Fig. 7. Ten videos were randomly selected from the ViML-Test dataset. They were rated on a scale of 0 to 10 based on general impressions, including aspects such as correctness, level of detail, richness, and fluency. The results of more than 100 sets of samples indicate that the manually adjusted prompts rating of 8.12 outperforms the auto-caption pipeline, while our merged captions achieve the second-best performance of 6.62. Despite being short and straightforward, object/background labels achieve a 6.02 evaluation score, demonstrating more correctness than other captions. Frame caption, music caption, and LLaVA caption obtain 4.3, 4.6, and 5.4, respectively, and these findings demonstrate the effectiveness of our captions and highlight the quality of our labeled captions by human annotators.

## 4.2 VIDEO GENERATION

We finetune the VideoCrafter-2.0 (Chen et al., 2024a) on ViML-Subset and WebVid (Bain et al., 2021a) using 8 Tesla-H800 GPUs with a batch size of 3 for 10,000 steps at a learning rate of 6e-6. The training data was randomly sampled from the full set, using the video captions as input. The evaluation results, shown in Table 3, include nine matrices on the VBench (Huang et al., 2024), indicating that fine-tuning the model on the ViML-3M dataset led to improvements of 1.38 in motion smoothness and 1.68 in subject consistency, with an overall performance boost(1.16 higher) compared to the VideoCrafter-2.0 checkpoint fine-tune on WebVid (Bain et al., 2021a). Visual examples of the generated content are provided in Fig. 8. This thorough evaluation and comparison of the tuned model's performance on critical metrics provides valuable insights into the effectiveness of the fine-tuning process and the potential benefits of leveraging the ViML-3M dataset for video generation tasks.

Table 4: Comparison of Video-LLaMA model performance on the extra test set MSRVTT (Xu et al., 2016). We compared our caption with LLaVA-OneVision-recaption (Li et al., 2024) data and fine-tuned the Video-LLaMA with the same setting. The result also shows that the caption quality of ViML is better.

| Model | Data | BLEU-4↑ | M↑ | ROGUE-L↑ | CIDEr↑ | BERT↑ |
|-------|------|---------|-----|----------|--------|-------|
| Video-LLaMA | Raw(2.5M Vid+395k Image) | 5.8 | 15.9 | 30.0 | 14.3 | 84.5 |
| Video-LLaMA | Raw + 0.3M(LLaVA-OneVision) | 6.2 | 18.5 | 30.1 | 13.5 | 85.6 |
| Video-LLaMA | Raw + 0.3M (ViML) | 8.1 | 19.7 | 32.1 | 14.5 | 85.9 |

Table 5: Comparison of Video-LLaMA model performance on the ViML-Test dataset. The table shows the results of three different versions of the Video-LLaMA model across five evaluation metrics, and the Video-LLaMA(ViML) version performs better on most evaluation indicators.

| Model | BLEU-4↑ | M↑ | ROGUE-L↑ | CIDEr↑ | BERT↑ |
|-------|---------|-----|----------|--------|-------|
| Video-LLaMA(Pretrain) | 0.5 | 4.6 | 11.6 | 0.09 | 84.4 |
| Video-LLaMA(Finetune) | 3.9 | 14.1 | 22.7 | 2.45 | 85.5 |
| Video-LLaMA(ViML) | 5.6 | 13.8 | 24.9 | 24.8 | 87.2 |

## 4.3 VIDEO UNDERSTANDING

**Experiment Setting** To evaluate the capability of our dataset in multimodal video understanding, we choose Video-LLaMA (Zhang et al., 2023a) as the baseline for the video captioning task. We use same model and training config as Video-LLaMA, which useVicuna-v0-7B as llama model (Zheng et al., 2023), ViT (Dosovitskiy et al., 2021) and Q-Former (Zhang et al., 2023b) as the video encoder and the linear projection layer from MiniGPT-4 (Zhu et al., 2023a). We train 4 epochs by ViML-3M, each containing 2500 iters with batch size 32. We compare it with two official model weights: the pre-train Video-LlaMA weight on WebVid (2.5M video-caption pairs) and the fine-tuned Video-LlaMA.

**Evaluation Metric** We evaluate video understanding models on the ViML-Test. As for the evaluation metric, we choose the commonly used metrics in text generation tasks-BLEU-4 (Papineni et al., 2002), ROGUE-L (Lin & Och, 2004), METEOR (Banerjee & Lavie, 2005), and CIDEr (Vedantam et al., 2015) to evaluate our result. All the metrics are computed using the pycocoevalcap (Lin et al., 2015) package. We also use BERTScore (Zhang et al., 2020) to evaluate the contextual similarity for each token in the ground truth and the predicted captions. The results are reported in Table 4 and Table 5. The official weights show relatively low performance, highlighting the challenge of ViML, and the data distribution differs from their training data.

In addition, we also evaluated three checkpoints from Video-LLaMA (Zhang et al., 2023a) by human evaluation in Fig. 7 and found that the Video-LLaMA-ViML evaluation result slightly lags behind Video-LLaMA-Finetune but performs significantly better than Video-LLaMA-Pretrain. We provide further details in Section 4.3 for a more comprehensive understanding of our model.

## 4.4 MUSIC GENERATION

We used text-to-music generation to evaluate the effectiveness of the video-music pair data and the labeled video caption and music caption. We use MusicGen (Copet et al., 2024) to generate music based on our video caption (VideoCap2Music) and music caption (MusicCap2Music). We use Kullback-Leibler Divergence (KL), Inception score (ISc), Frechet distance (FD), and Frechet Audio Distance (FAD) (Kilgour et al., 2018) to evaluate the generated music. Besides, we use the ImageBind-AV score (IB) to evaluate the audio-visual alignment between the video and the generated music.

For the model with text input in Tab. 6, compared with video caption, the evaluation results on music caption are 0.13 better in KL, 0.65 in ISc, 3.64 in FD, and 1.21 in FAD, showing the domain gap of multimodal descriptions. We also conduct extended experiments on video-to-music generation by using VidMuse (Tian et al., 2024), a state-of-the-art model. The results in the table show the high audio quality and strong audio-visual alignment between the video and the generated music

Table 6: Music generation evaluation results on the ViML-Test. We compare two types of captions and their 5 metrics. The results show that music captions perform better in music generation tasks.

| Method | Input | KL↓ | ISc↑ | FD↓ | FAD↓ | IB↑ |
|---|---|---|---|---|---|---|
| VideoCap2Music | Text | 3.22 | 1.79 | 57.17 | 15.04 | 0.09 |
| MusicCap2Music | Text | 3.10 | 2.44 | 53.53 | 13.83 | 0.14 |
| VidMuse (Tian et al., 2024) | Video | 0.99 | 1.23 | 48.14 | 5.08 | 0.18 |

achieved by our dataset. This comparison shows that there is still a significant research gap between caption-music-video, and using video as a music generation condition is a highly potential approach.

## 5 CONCLUSION

We introduce ViML, a comprehensive and accurate multi-modality visual-audio dataset to address the dataset gap. By utilizing the inherent value of trailers, which integrate visual, audio, and contextual elements, ViML offers detailed and precise multi-modality annotations. Our systematic captioning framework adaptively merges visual and musical perspectives, ensuring that the annotations capture the richness of multimodal content. Experimental results demonstrate the high quality of the ViML dataset, its effectiveness for fine-grained multimodal-language model training, and a variety of down-stream applications. We believe this innovative dataset will unlock new possibilities in video content generation and significantly advance research in visual-audio understanding. The comprehensive and diverse nature of ViML makes it a valuable asset for the research community, paving the way for novel applications that leverage the power of multimodal learning.

## 6 ETHICAL ISSUES

Our dataset adheres to the latest public video-audio-text dataset standards and involves a rigorous pipeline for data processing, ensuring its suitability for research purposes. We will explain the following aspects of our work license.

**Licensing and access**: ViML is a Research-Only dataset, and we refer to the Creative Commons(CC BY-NC-SA 4.0)(https://creativecommons.org/share-your-work/cclicenses/) to release our metadata, same as InternVid Wang et al. (2023c).

**Ethics and responsible use**: For the development of the community and more strict license requirements, the users of ViML must also follow the Research Use of Data Agreement v1.0.(License) same as Panda-70M (Chen et al., 2024b).

**Consent and privacy**: We provide text descriptions only. In ViML, we only provide text descriptions, excluding any sensitive information. Raw videos and metadata containing personal or privacy-sensitive content are not included. During data processing, all text descriptions are generated from an existing open-source model, ensuring they are highly compressed and cannot reconstruct the original video information. Therefore, users cannot recover raw data or any other sensitive information from the text descriptions.

**Legal data collection steps** We use YouTube's official API to ensure compliance with the platform's terms of service and data usage policy. We filter the metadata by (1)Tag Filtering: Use YouTube video tags to exclude videos that do not meet research purposes or contain sensitive content. (2)Content Classification: Automatically filter out inappropriate categories (e.g., adult content, political content, 18+). (3)User-Provided Tags: Combine with user-provided tags to identify and filter potential sensitive content.

**Test Set and Benchmark**: Since we need to release the video of the testing set, we are filtering the ViML-Test set that is strictly followed by the CC(https://creativecommons.org/licenses/by/). We further applied manual checks to ensure the test set included suitable information only.

Generally, our dataset is meticulously curated during collection, adhering to the latest public video-audio-text dataset standards, ensuring its suitability for research purposes. We remain vigilant about the rapidly evolving license requirements and are committed to continuously updating and further filtering the data to meet these needs.

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

## A   CAPTIONING PIPELINE

### A.1   MUSIC CAPTION

As trailer videos often contain a cacophony of audio tracks, which typically include background music and vocals, captioning audio poses a significant challenge. Therefore, to achieve optimal music captioning results, we initially utilize Demucs (Rouard et al., 2023) for vocal separation on each audio clip. Subsequently, LP-MusicCap (Doh et al., 2023a) is leveraged to caption the resultant audio, without vocals, from the separation process. More results and comparisons of music captions are shown in Fig. 9.

## B   CAPTIONING PIPELINE

### B.1   FRAME CAPTION

We utilize LLava-v1.6-vicuna-7B as the image understanding model to generate captions for video clips. Initially, we filter out clips with durations of less than 1.0 seconds. Subsequently, we sample three frames at the fractions of 0.2, 0.5, and 0.8 of each clip's total duration. The prompt is as follows:

```
Please describe the image in detail.
```

**Music Caption for Unseparated Audio**: This audio contains a male voice speaking in a lower key. Then a spoken word recording starts playing a melody on a marimba. This is an amateur recording. This may be playing in a tutorial video on the djembe.

**Music Caption for Separated Audio**: This audio contains someone playing a xylophone sound and rattles. This is an amateur recording. You can hear clicking and recording noises. T This audio contains a male voice speaking in a lower key. Then a spoken word recording starts playing a melody on a marimba. This is an amateur recording. This may be playing in a tutorial video on the djembe. his song may be playing demonstrating specific sounds on a device.

(a)

**Music Caption for Unseparated Audio**: The low quality recording features a flat male vocal talking, after which there is a synth pad speaking in the background. The recording is noisy and in mono.

**Music Caption for Separated Audio**: The low quality recording features a suspenseful synth pad played over playback that consists of loud bell tones and some sea waves sounds. The recording is noisy and in mono.

(b)

**Music Caption for Unseparated Audio**: The low quality recording features a flat female vocal talking over playback instrumental that consists of a flat male vocal talking, after which there is a harmonizing female vocal melody. The recording is noisy and in mono.

**Music Caption for Separated Audio**: This is the type of horn that would be heard in a distant battlecry. The clip features just this war horn, which sounds like the sound of a dog barking.

(c)

**Music Caption for Unseparated Audio**: This music is instrumental. The tempo is medium with a male voice speaking in an instructive manner. The music is like a tutorial on the guitar.

**Music Caption for Separated Audio**: This audio contains someone playing a marimba melody on a horn. This is an amateur recording. This may be playing in a church.

(d)

Figure 9: We provided an extra comparison of the music caption before and after the track separation. With our separation, the caption includes the description of the human voice as highlighted in red.

## B.2 LLAVA-VIDEO CAPTION

We use the image understanding model LLaVA-13b (Liu et al., 2024) to caption each of our video clips. Specifically, for each clip, we sample frames at positions 0.1, 0.3, 0.5, 0.7, and 0.9, and then horizontally concatenate them into a single image, which serves as input for LLaVA-13b to generate captions. We construct the caption prompt as follows:

```
These are some keyframes of a video.
Please use one sentence to summarize the content of the video
    in detail.
Summarize the content of the entire video but not describe
    keyframes frame by frame.
```

Moreover, we observe that the results of LLaVA captioning often contained some redundant information, as illustrated by the green sections in Figure Fig. 11. Therefore, we use LLaMA-13b to refine the LLaVA captions, eliminating much of the extraneous content and rendering the final captions more in line with human expression. We construct the prompt as follows:

```
[{caption}] This is a description of a video.
Please polish it to an overall video description in one
    sentence and give me only the content of the video.
Do not use the words 'frame' and 'video'.
```

```
    Describe the content of the video directly, which means do not
        start with 'The video...' or something like that.
    Do not add extra information that is not included in the
        original description.
    Here is an example: A dancer in a vibrant orange skirt and gray
         jacket moves gracefully across the stage, her movements
        fluid and expressive.
```

By combining frames and secondary polishing, we finally obtain high-quality captions that contain the main content information of the clips.

### B.3 MERGED CAPTION

We use LlaMA-13B to merge the multiple captions, by constricting a pre-designed caption prompt as follows:

```
    There are some descriptions of a video,
    including video caption, music caption, background caption and
        the main objects in the video.
    Please combine all the descriptions into an overall description
         of the video in only one paragraph.
    Finally, please rewrite and polish it into an overall video
        description in one paragraph and give me only the content
        of the video.
        Background: {background}
        Main objects: {objects}
        Video caption 1: {image_cap}
        Video caption 2: {frame_cap}
        Music caption: {music_cap}
```

As illustrated in Figure Fig. 10, we use the LLaMA-13B to merge frame captions, lava captions, music captions, objects, and background into a final merge caption. By merging multiple captions, the merge caption offers a comprehensive and detailed description of the audiovisual content of videos. Its components mutually complement one another, ensuring that every aspect of the narrative receives attention and providing unique insights into various facets of the video content.

## C DATASET DETAILS

Here is one metadata example of our dataset. We introduce the basic information of video and clips in the "basic" tag, including their duration, quality evaluation score, etc. The useful caption and description are saved in the "scene" tag.

```
{
    "basic": {
        "video_id": "-3r7ptfObEs",
        "video_path": "group_33/-3r7ptfObEs.mp4",
        "video_duration": 71.73333333333333,
        "video_resolution": [
            720,
            1280
        ],
        "video_fps": 30.0,
        "clip_id": "-3r7ptfObEs_0000000",
        "clip_path": "video_dataset_33/-3r7ptfObEs_0000000.mp4",
        "clip_duration": 7.033333333333333,
        "clip_start_end_idx": [
            0,
            211
```

```
        ],
        "imaging_quality": 36.83453941345215,
        "of_score": 12.92151,
        "aesthetic_score": [
            3.010026454925537,
            3.664743423461914,
            3.994750499725342
        ]
    },
    "camera": {
        "view_scale": "",
        "movement": "",
        "speed": ""
    },
    "misc": {
        "frame_caption": [
            "a person standing in a room with a laptop on their lap.
                ",
            "a black and blue background with the name of the series.
                ",
            "a black and white image of the words manga and comics. "
        ],
        "music_caption": [
            {
                "text": "This is an indie rock music piece. There is a
                    male vocalist singing melodically in the lead.
                    The main tune is being played by the electric
                    guitar while the bass guitar is playing in the
                    background. The rhythm is provided by a simple
                    acoustic drum beat. The atmosphere is easygoing.
                    This piece could be used in the soundtrack of a
                    teenage drama TV series as the opening theme.",
                "time": "0:00-10:00"
            }
        ]
    },
    "scene": {
        "objects": [
            " Matheus Trindade",
            " de",
            " original",
            "Series",
            " Manga & Comics"
        ],
        "background": "Dark",
        "ocr_score": 0.07147466104497355,
        "caption": "The video is a series of original de Matheus
            Trindade comics, showcasing the artist's unique style
            and storytelling.",
        "polish_caption": "Original de Matheus Trindade comics
            showcase the artist's unique style and storytelling,
            featuring a series of vibrant and expressive
            illustrations that bring characters and scenes to life.",

        "merge_caption": "In this captivating video, Matheus
            Trindade's original comics come to life with vibrant and
             expressive illustrations, showcasing the artist's
            unique style and storytelling. Set against a dark
            background, the video features a series of dynamic and
```

```
            colorful images, including a person sitting in a room
            with a laptop, a black and blue background with the name
             of the series, and a black and white image of the words
             \"manga\" and \"comics.\" The indie rock music piece,
            featuring a male vocalist and simple acoustic drum beat,
             adds an easygoing atmosphere to the video, making it
            perfect for the opening theme of a teenage drama TV
            series."
    }
}
```

**Frame Caption**: "a man in a suit and tie sitting next to another man in a suit and tie. ", "a man in a suit sitting in front of another man in a suit. ", "a man in a suit and tie sitting at a table with another man. "

**LLaVA Caption**: "A professional man in a suit listens intently to a standing colleague, who gestures with his hands as he speaks, in a serious and focused business environment."

**Music Caption**: "The low quality recording features a flat male vocal talking, after which there is a synth pad playing. It sounds like a tutorial and the recording is noisy and in mono."

**Objects**: " tie", "suit", "chair", "Man", "desk"
**Background**: "Office"

**Merge Caption**: "In a professional office setting, a well-dressed man in a suit and tie listens intently to his standing colleague, who passionately gestures with his hands as he speaks. The scene is set against a backdrop of sleek, modern furniture and a tidy desk, creating a focused and serious atmosphere. The video features a series of shots showcasing the men in suits engaging in a meeting or discussion, with one man sitting next to another, or sitting in front of each other at a table. The low-quality recording features a flat male vocal and a synth pad, giving the video a tutorial-like feel."

(a)

**Frame Caption**: "a woman sitting at a table with a glass of wine. ", "a woman sitting at a table talking on a phone. ", "a woman sitting at a table talking on the phone. "

**LLaVA Caption**: "A woman sits at a table, surrounded by wine glasses, bottles, and a cake, engaging in conversation and interacting playfully with the objects on the table. "

**Music Caption**: "This clip features a female voice speaking in an instructive manner. In the background you can hear birds chirping. This is an amateur recording. "

**Objects**: "wine glass", "table", "kitchen", "Woman", "candle"
**Background**: "Kitchen"

**Merge Caption**: "In this video, a woman sits at a table in a cozy kitchen surrounded by wine glasses, bottles, and a cake, engaging in lively conversation and playfully interacting with the objects on the table. The scene is set against a peaceful background of birds chirping, adding to the warm and inviting atmosphere. The woman is shown speaking on the phone, laughing and smiling as she enjoys her wine and the company of those around her. The video features an amateur recording of a female voice speaking in an instructive manner, adding to the sense of intimacy and authenticity. "

(b)

**Frame Caption**: "a man with long hair is looking at a mirror. ", "a man with long black hair talking to a woman in front of a mirror. ", "a man with long hair is looking at a woman. "

**LLaVA Caption**: "A man with a beard expresses a range of emotions and reactions, from thoughtfulness to amusement, as he engages in a conversation or interview. "

**Music Caption**: "The low quality recording features a tutorial that consists of a flat male vocal talking over sustained strings melody. It sounds like a tutorial and the recording is noisy and in mono."

**Objects**: " door", " room", " television", "Man", " beard"
**Background**: "Doorway"

**Merge Caption**: "The video, set against a doorway background, features a man with a beard engaging in a conversation or interview, expressing a range of emotions and reactions from thoughtfulness to amusement. He is surrounded by elements of a room, including a television, and is occasionally joined by a man with long hair who looks at a mirror or talks to a woman. The low-quality recording includes a tutorial with a flat male vocal over sustained strings melody, giving the video a noisy and mono feel. "

(c)

Figure 10: We demonstrate more examples of the merged captions. As shown in the examples, all key information from different captions is merged together into a fluent paragraph.

Video Clip Input

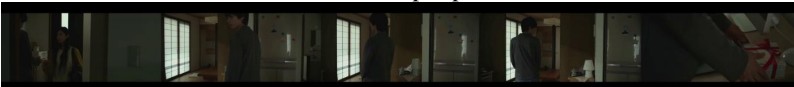

LLaVA Caption

The video appears to be a series of close-up shots of a man in a blue sweater, possibly in a conversation or interview setting, with the focus on his facial expressions and reactions.

Polished Caption

A man in a blue sweater expresses a range of emotions, from thoughtfulness to amusement, as he engages in a conversation or interview, with his facial expressions and reactions taking center stage.

Video Clip Input

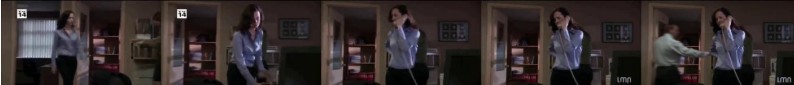

LLaVA Caption

The video appears to be a montage of various scenes, possibly from a movie or TV show, featuring a person walking through a room with multiple windows, and another person standing in front of a refrigerator. The overall content of the video is not clear without additional context.

Polished Caption

A person walks through a room with multiple windows, while another person stands in front of a refrigerator, creating a sense of familiarity and everyday life.

Video Clip Input

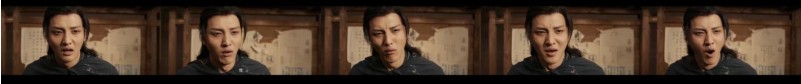

LLaVA Caption

The video appears to be a compilation of scenes from a television show or movie, featuring a woman in various positions and actions, such as walking, talking on the phone, and interacting with other characters.

Polished Caption

A confident and stylish woman navigates various situations and interactions, showcasing her grace and poise as she walks, talks on the phone, and engages with other characters.

Video Clip Input

LLaVA Caption

The video appears to be a close-up montage of a person with a surprised or shocked expression, possibly reacting to something unexpected or dramatic.

Polished Caption

A person with a surprised or shocked expression reacts to something unexpected or dramatic, their face contorting into a mixture of fear and disbelief.

Figure 11: While using LLaVA as the backbone model for our video caption, we find that it contains a lot of "Oral habit", as highlighted in green text in this figure. We further apply a language model to reception the sentence.