# OpenReview forum: "ViML: A Video, Music, Language Unified Dataset for Understanding and Generation"
_ICLR.cc/2025/Conference — ICLR 2025 Conference Withdrawn Submission_

### Official Review · Reviewer_RUAQ · 2024-10-28

**Soundness:** 2
**Presentation:** 3
**Contribution:** 2
**Rating:** 5
**Confidence:** 3

**Summary:**

The paper introduces ViML, a large-scale multi-modality-to-language dataset designed to capture the inherent audio-visual correlations, which are often neglected in existing datasets that treat visual and audio as separate tasks. Focusing on trailers, ViML getting plenty of video clips with high-quality background music and sound effects and annotate them with frame captions, music captions, object and background descriptions, and merged captions.

**Strengths:**

- The paper employs a highly detailed set of evaluation scores to assess the information content of audio-visual segments from multiple perspectives, and it provides intuitive examples to illustrate the meaning of these scores.
- The dataset achieves superior performance across a variety of downstream task indicators.

**Weaknesses:**

1. Although the paper conducts a multifaceted assessment of the quality of video imagery and background music, the evaluation of sound effects in terms of quality appears to be omitted. In many instances, it is the sound effects, rather than the background music, that have a direct correlation with the content of the video.

2. I have noticed that in the evaluation pipeline, when assessing the alignment of audio and video, the paper seems not to separate background music from sound effects. Since the trainset of ImageBind is Audioset, which does not distinguish between music and sound effects, the simultaneous presence of background music and sound effects is likely to impact the Audio-Vision similarity scores.

**Questions:**

Please refer to weakness.

---

### Official Review · Reviewer_dxfg · 2024-11-02

**Soundness:** 3
**Presentation:** 3
**Contribution:** 3
**Rating:** 5
**Confidence:** 4

**Summary:**

This paper introduces ViML, a large-scale multimodal dataset and framework focused on video trailers, designed to address limitations in existing video-language datasets. Here are the main contributions:
- The paper introduces ViML, a large-scale multimodal dataset based on video trailers that uniquely incorporates both visual and audio content, containing over 20M video clips from 290k trailers.
- The authors develop a multimodal captioning pipeline that combines multiple state-of-the-art models and includes a language model fusion strategy to generate detailed captions.
- They create two specific dataset versions: ViML-3M with 3 million annotated samples and ViML-Test with manually-adjusted captions for evaluation purposes.
- The paper provides comprehensive evaluation through benchmarking, human evaluation, and model fine-tuning experiments to demonstrate the dataset's quality and utility for various multimodal tasks.

**Strengths:**

- Uniquely focuses on audio-visual correlations through paired video clips and multimodal captions, addressing a gap in traditional datasets.
- Employs a sophisticated captioning pipeline combining state-of-the-art models with language model fusion to produce detailed, accurate multimodal annotations.
-Features diverse content (films, news, gaming) with custom background music, enabling fine-grained multimodal model training. Includes comprehensive evaluation metrics and benchmarks to validate dataset quality.

**Weaknesses:**

- The empirical evidence supporting the method's effectiveness appears limited. In particular, the results presented in Tables 4 and 5 show only marginal performance improvements, raising questions about the practical value of the proposed approach.
- The experimental validation could benefit from a more comprehensive evaluation framework, the study would be strengthened by:
  - Including additional benchmark datasets (e.g., ShareGPT4Video) for broader validation
  - Conducting more diverse understanding tasks
- There are concerns regarding potential information loss in the synthetic caption generation process. The current approach may not adequately preserve the original data's information.

**Questions:**

As described in the weakness section

---

### Official Review · Reviewer_y5px · 2024-11-03

**Soundness:** 2
**Presentation:** 3
**Contribution:** 2
**Rating:** 5
**Confidence:** 4

**Summary:**

In this paper, the authors introduce a new multimodal captioning dataset ViML. The dataset contains 20M video clips from trailers of various types such as movies, documentaries, games, etc. The trailer videos are chopped into smaller clips. Then multiple captions such as frame captions, video captions, and music captions are generated for each clip using various LLM-based models. The authors also generate labels for objects and the background category of the clip. They combine all these captions and object categories to produce a complete multimodal caption using the llama-2 model. The authors also create a subset with 3M clips and a 1k clips test set with manually adjusted final captions. The authors evaluate the dataset on tasks such as video captioning, video generation, and music generation to show the complexity of the dataset.

**Strengths:**

1. I agree with the authors that most existing captioning datasets focus on a single modality or only use the other modality as an aid. To my knowledge, this is the first multimodal captioning dataset that includes musical captions along with captions for the video.
2. The large scale of the dataset with 20M clips and 3M high-quality subset clips is good for the development of multimodal models.

**Weaknesses:**

1. One of the important contributions of this dataset is the musical captions given to each clip. However, the statistics provided for the dataset say 40% of the clips are less than 1 second and 72% of the clips are less than 3 seconds. I am not confident about the information, quality, and correctness of the music captions for such short-duration clips.

2. Another weakness is that the entire pipeline of caption generation and merging is done with various LLMs. There are no proper checks done to evaluate the hallucination effects that keep accumulating over multiple steps in the captioning pipeline. The human evaluation is only done for a small number of manually adjusted captions. It is unclear how well the various LLMs in the pipeline were correctly describing without hallucinating, especially for very short-duration music captions.

3. Since the entire pipeline is LLM automated, it would have been useful to discuss more the effects each one of them had on the final merged caption.

**Questions:**

1. The authors claim in section 3.1 that trailers are vague and incoherent, music and voice-over are commentaries rather than produced by characters. This means that the music and voice-over are not necessarily aligned with the video. But then the authors use the imagebind model to assess the semantic alignment between these clips. Do the music and voice-over commentaries not affect this alignment score? Did the authors consider/evaluate the effect of this? This is unclear.

2. As discussed earlier, I would like to get the views of the authors regarding music captions for less than 1s clips. Is this not too little information for a model to generate a meaningful musical caption?

3. For the manually adjusted test set creation, the input for this step seems to be the merged caption produced by llama2 which again aggregated various captions produced by other LLMs in previous steps. So if any of the previous models had hallucinations, then how do the authors guarantee that the test set captions are ground truth? Is there anything done to verify this? Does the human evaluation verify this?

4. In section 4.1 "Despite being short and straight-forward, object/background labels achieve a 6.02 evaluation score, demonstrating more correctness than other captions". What does this imply? Why a frame-level caption or music caption is a poor description of the video rather than simply the object/background labels? How do these findings highlight the effectiveness of the captions as claimed by the authors? This is unclear to me.

5. In Table 5, why is the BERT similarity alone high and comparable for the pre-trained video-llama with the fine-tuned version and the video-llama trained on ViML? Does the vast difference between the metrics of pre-trained llama and the other two versions indicate difficulty/domain mismatch in the videos or the caption style is very different?

---

### Official Review · Reviewer_Qes1 · 2024-11-04

**Soundness:** 3
**Presentation:** 3
**Contribution:** 1
**Rating:** 3
**Confidence:** 5

**Summary:**

This paper proposes ViML, a large-scale multi-modality dataset of 3M video-caption pairs. A systemic captioning framework is proposed to generate music, video, and frame captions separately and a multimodal caption is then merged by uni-modal captions along with objects and background labels.

**Strengths:**

1. Large-scale music caption datasets are scarce. A large-scale video-language dataset with high-quality music narration benefits the community. The proposed dataset also features high image quality, aesthetic scores, and motion characteristics, which shows promising potential for training high-quality text-to-video/text-to-audio&video generation models.

2. The data curation pipeline looks sound and intuitive. The data filtering strategy has proved to be effective through ablations.

3. Experiments on several downstream tasks reveal the high-quality nature of the proposed dataset.

**Weaknesses:**

1. The entire caption pipeline is trivial and not inspiring. Several similar data curation pipelines have been proposed [1-2], and the authors selected the existing MusicCaps model as the music captioner. Therefore, the data pipeline is more likely to be an integration of several existing uni-modal captioners, and even the integration tool(an LLM) has been used by several previous methods, making the technical contribution insufficient.

2. The manuscript lacks ablations. Why select Coca as the image captioner and LLaVA as the video captioner? Can LLaVa be applied to generate image captions since its performance is better than CoCa? Why not use video-encoder-based MLLM as the video captioner such as VideoChat [3] and Video-LLaVa [4]? More clarifications are needed.

3. Experiments are inadequate. For video generation, the authors selected WebVid as the baseline, which is a relatively earlier dataset. How about the comparison with some newer datasets such as LVD-2M [5] or Panda-70M [6]? Besides, VBench has 16 dimensions, yet the authors only report 9, what is the rationale? For caption, the authors should test more caption datasets such as Valor32k [7], FAVD [8], and DiDeMo [9] that include rich audio-visual content.

Reference:
[1] Chen, Sihan, et al. "Vast: A vision-audio-subtitle-text omni-modality foundation model and dataset." Advances in Neural Information Processing Systems 36 (2023): 72842-72866.
[2] Wang, Yi, et al. "Internvideo2: Scaling video foundation models for multimodal video understanding." arXiv preprint arXiv:2403.15377 (2024).
[3] Li, KunChang, et al. "Videochat: Chat-centric video understanding." arXiv preprint arXiv:2305.06355 (2023).
[4] Lin, Bin, et al. "Video-llava: Learning united visual representation by alignment before projection." arXiv preprint arXiv:2311.10122 (2023).
[5] Xiong, Tianwei, et al. "LVD-2M: A Long-take Video Dataset with Temporally Dense Captions." arXiv preprint arXiv:2410.10816 (2024).
[6] Chen, Tsai-Shien, et al. "Panda-70m: Captioning 70m videos with multiple cross-modality teachers." Proceedings of the IEEE/CVF Conference on Computer Vision and Pattern Recognition. 2024.
[7] Chen, Sihan, et al. "Valor: Vision-audio-language omni-perception pretraining model and dataset." arXiv preprint arXiv:2304.08345 (2023).
[8] Shen, Xuyang, et al. "Fine-grained audible video description." Proceedings of the IEEE/CVF Conference on Computer Vision and Pattern Recognition. 2023.
[9] Anne Hendricks, Lisa, et al. "Localizing moments in video with natural language." Proceedings of the IEEE international conference on computer vision. 2017.

**Questions:**

My major concerns are listed in the weaknesses part above. I only have minor questions here.

1. What is the data source? Are all videos selected from YouTube? Will the data source be overlapped with some existing datasets, such as HD-Vila and YT-Temporal-1b?

2. The average length is 4.6s, which is a relatively short length compared with other datasets. Why not adopt the stitching procedure in Panda-70M to make the split video longer?

3. I believe the proposed dataset is a perfect data source for sounding video generation. Have the authors tested this capability using some SVG model, such as MM-Diffusion [1] or MM-LDM [2]?

4. For music generation capabilities, are there any comparable results? I believe the text-to-music models and video-to-music models should achieve better results after training on the proposed dataset.

Reference:
[1] Ruan, Ludan, et al. "Mm-diffusion: Learning multi-modal diffusion models for joint audio and video generation." Proceedings of the IEEE/CVF Conference on Computer Vision and Pattern Recognition. 2023.
[2] Sun, Mingzhen, et al. "MM-LDM: Multi-Modal Latent Diffusion Model for Sounding Video Generation." arXiv preprint arXiv:2410.01594 (2024).

---

### Note · Authors · 2024-11-14

I have read and agree with the venue's withdrawal policy on behalf of myself and my co-authors.